

# Drought and Flood in the Anthropocene:
# Modelling Feedback Mechanisms

Giuliano Di Baldassarre[1,2], Fabian Martinez[1], Zahra Kalantari[3,4], Alberto Viglione[5]

[1]Uppsala University, Department of Earth Sciences, 75236 Uppsala, Sweden
[2]Centre for Natural Disaster Science (CNDS), 75236 Uppsala, Sweden
[3]Stockholm University, Department of Physical Geography, 106 91 Stockholm, Sweden
[4]Bolin Centre for Climate Research, 106 91 Stockholm, Sweden
[5]Vienna University of Technology, Centre for Water Resource Systems, 1040 Vienna, Austria

*Correspondence to*: Giuliano Di Baldassarre (giuliano.dibaldassarre@geo.uu.se)

**Abstract.** Over the last few decades, numerous studies have investigated human impacts on drought and flood events, while conversely other studies have explored human responses to hydrological extremes. Yet, there is still little understanding about the dynamics resulting from their interplay, i.e. both impacts and responses. Current quantitative methods therefore fail to assess future risk dynamics and, as a result, while risk reduction strategies built on these methods often work in the short term, they can lead to unintended consequences in the longer term. In this paper, we review the puzzles and dynamics resulting from the interplay of society and hydrological extremes, and describe our initial efforts to model hydrological extremes in the Anthropocene. In particular, we first discuss the need for a novel approach to explicitly account for human interactions with both drought and flood events, and then present a stylized model simulating the reciprocal effects between water management and hydrological extremes. Lastly, we highlight the unprecedented opportunity offered by the current proliferation of big data to unravel the coevolution of hydrological extremes and society across scales, and along gradients of social and hydrological conditions.





## 1 Introduction

Throughout history, human societies have been severely impacted by hydrological extremes, i.e. drought and flood events. The collapse of various ancient civilizations, for instance, has been attributed to the occurrence of hydrological extremes (e.g. Munoz et al., 2015). Fatalities and economic losses caused by drought and flood events have dramatically increased in many regions of the world over the past decades (Di Baldassarre et al., 2010; Winsemius et al., 2015) and, currently, more than 100 million people per year are affected by hydrological extremes (UN-ISDR, 2016). There is serious concern about future hydrological risk (broadly defined here as a combination of hazard, vulnerability and exposure) given the potentially negative impact of climatic and socio-economic changes (Hallegatte et al., 2013; Jongman et al., 2014; IPCC, 2014). Thus, it is essential to realistically capture where, how and why risk will plausibly change in the coming decades, and develop appropriate policies to reduce the negative impacts of hydrological extremes, e.g. economic losses and fatalities, while retaining the benefits of hydrological variability, e.g. supporting biodiversity and ecosystem functions.

Human societies have (intentionally or accidentally) altered the frequency, magnitude and spatial distribution of flood and drought events (Falkenmark and Rockström, 2008; Di Baldassarre et al., 2009; Vörösmarty et al., 2013; Montanari et al., 2013; AghaKouchak et al., 2015). Dams and reservoirs are examples of water management measures that deliberately change hydrological variability (Ye et al., 2003) and significantly affect hydrological extremes, as schematically depicted in Figure 1.

Moreover, water abstraction and irrigation have a significant impact on the occurrence of hydrological drought (Van Loon et al., 2016); while flood protection measures, such as levees, alter the frequency, magnitude and spatial distribution of flood events (Di Baldassarre et al., 2009; Blöschl et al., 2013; Heine and Pinter, 2012). Hydrological extremes can also be affected by other human activities such as land-use change, deforestation, urbanisation, drainage of wetlands and agricultural practices (Kalantari et al., 2014; Savenije et al., 2014; Destouni et al., 2015).

While human societies shape hydrological extremes, hydrological extremes in turn shape human societies. Following the impact of drought or flood events, humans respond and adapt to hydrological extremes through a combination of spontaneous processes and deliberate strategies that can lead to changes in social contracts (Adger et al., 2013). Adaptive responses can take place at the individual, community or





institutional level (Myers et al., 2008; Penning-Rowsell et al., 2013). Early warning systems, risk awareness programs, and changes of land-use planning are examples of adaptive responses that often occur at the local or central government level following hydrological extremes (Pahl-Wostl et al., 2013). Moreover, structural risk reduction measures, such as reservoirs or levees, are also planned, implemented

or revised after the occurrence of drought or flood events, and they in turn (again) change the frequency, magnitude and spatial distribution of hydrological extremes (Di Baldassarre et al., 2013a).

In the recent decades, natural and engineering scientists have analysed numerous facets of human impacts on drought and flood events, while conversely economists and social scientists have explored human responses to hydrological extremes. Yet, the dynamics resulting from the mutual shaping (i.e. both

impacts and responses) of hydrological extremes and societies are still not well understood. As a result, current quantitative methods fail to assess the dynamics of hydrological risk and, while risk reduction strategies built on these methods often work in the short term, they can lead to unintended consequences in the long term. To overcome this lack of knowledge, there has been increasing interest in socio-hydrology in the last few years (e.g. Sivapalan et al., 2012; Srinivasan et al., 2012; Di Baldassarre et al.,

2013b; Montanari et al., 2013; Viglione et al., 2014; Elshafey et al., 2014; van Emmerick et al., 2014; Sivapalan and Bloeschl, 2015, Loucks, 2015; Troy et al., 2015; Gober and Weather, 2015; Pande and Savenije, 2016; Blair and Buytaert, 2016), which aims to develop fundamental science underpinning integrated water resources management (IWRM). Socio-hydrology builds on a long tradition of studies exploring the interplay of nature and society and the implications for sustainability, including political

ecology, social-ecological systems, ecologic economics, complex system theories and research on planetary boundaries (Swyngedouw, 1999; Folke et al., 2005; Liu et al., 2007; Ostrom, 2009; Rockström et al., 2009; Kallis and Norgaard, 2010).

In this context, this paper describes the puzzles and dynamics emerging from the interplay of society and hydrological extremes, discusses the need for a novel approach to explicitly account for both drought and

flood events, and presents initial efforts to model hydrological extremes in the Anthropocene.





## 2 Emerging dynamics and puzzles

Various dynamics result from the interactions between human societies and hydrological extremes. Learning or adaptation effects emerge when more frequent events are associated with decreasing vulnerability (Di Baldassarre et al., 2015). This effect can be attributed to informal adaptive processes,

such as temporary and permanent migration, or changes in policies triggered by the occurrence of hydrological extremes (Pahl-Wostl et al., 2013). For instance, Mechler and Bouwer (2014) showed decreasing flood fatalities in Bangladesh over the past 40 years (Figure 2, left panel). This reduced vulnerability can be attributed to coping and adaptation capacities gained by individuals or communities after the experience of extreme events.

Societies are not only shaped by the occurrence of hydrological extremes, but also by the perception of current and future risk (Dessai and Sims, 2010). This can explain the emergence of what is termed here as the forgetting or levee effect, i.e. less frequent events associated with increasing vulnerability. Since White (1945), the literature has provided various examples that show that the negative impact of an extreme event tends to be greater if such an event occurs after a long period of calm. Prolonged absence

of drought or flood events can be caused by climatic factors (e.g. flood-poor periods; Hall et al., 2014) or the introduction of structural risk reduction measures, such as reservoirs (Figure 1). One example is the case of Brisbane, where the introduction of a flood retention reservoir in the 1970s has shaped risk perception in the local community, which perceived Brisbane as flood-proof until a catastrophic flood event occurred in 2011 (Bohensky and Leitch, 2014).

Learning and forgetting effects have been reported in different parts of the world in a variety of empirical studies, e.g. collection of case studies reported in Di Baldassarre et al. (2015). The emergence of these dynamics suggests the intuitive tendency that the impact of drought or flood events depend on whether their occurrence is expected or not. Yet, these dynamics have mainly been reported as narratives in specific case studies. It is still unclear whether they are exceptional cases or generic mechanisms, and

whether they occur randomly or within certain social and hydrological circumstances. This lack of knowledge prevents their explicit inclusion on the analytical tools that undertake a quantitative assessment of hydrological risk.




Besides the inability to capture learning and forgetting dynamics, traditional methods for risk assessment cannot explain interactions between floods, droughts and water management as they focus on either drought or flood hazard (e.g. Shahid and Behrawan, 2008; Jongman et al., 2014). For instance, while reservoirs theoretically alleviate both flood and drought events (Figure 1), reservoir operation rules

(Mateo et al., 2014) mitigating drought are different from the ones mitigating flood. To cope with drought, reservoirs are typically kept as full as possible, working as a buffer during low flow conditions, whereas to cope with flood, reservoirs are often kept as empty as possible, allowing the storage of a large quantity of water from extreme rainfall or rapid snow melt conditions. These reservoir operation rules can change over time depending on various factors, including whether the most recently experienced disaster was

caused by a drought or a flood event. As a result, the negative impact of flood events occurring immediately after a long period of drought conditions can be exacerbated.

For example, the aforementioned catastrophic 2011 flooding of Brisbane occurred after an exceptionally long, multi-year drought (the so-called "Millennium Drought"; Van Dijk et al., 2013), which triggered changes in reservoir management (van den Honert and John McAneney, 2013). In particular, operation

rules of the flood mitigation reservoir build in 1970s were changed, and the reservoir was used instead as a buffer to cope with drought conditions. This change in operation rules led to higher water levels in the reservoir, which was then less unable to store much water and alleviate the 2011 flood event. Meanwhile, paradoxically, the presence of the reservoir triggered the popular belief that Brisbane was flood proof and made the population more vulnerable. The combination of these events made the 2011 flooding a major

disaster (Bohensky and Leitch, 2014).

Research on climate change suggests that many regions around the world might experience, in the near future, alternate periods with prolonged drought conditions and extreme flood events (IPCC, 2014). Thus, it is vital to understand if (and how) human responses to drought events might exacerbate the impact of future floods, and vice versa.

Furthermore, a focus on either drought or flood events can limit the interpretation of the role of global drivers of hydrological risk, such as climatic and socio-economic changes. For example, a number of recent studies (e.g. Di Baldassarre et al., 2010; Winsemius et al., 2015) have shown that socio-economic changes have been the main driver of increasing flood risk in Africa, while climate has (so far) played a





smaller role. Yet, by focusing on flood risk alone, these studies did not consider the hypothesis that climate may have led to longer and more severe drought conditions, which in turn have enhanced the need for individuals and communities to move closer to rivers, thus leading to greater exposure to flooding. Thus, it is still largely unexplored how sequences of drought and flood events make a difference in the

dynamics of hydrological risk. This puzzle requires further research on the mutual shaping of human societies and hydrological extremes, to which this paper aims to contribute.

## 3 Hydrological extremes in the Anthropocene

To reveal the aforementioned dynamics resulting from the mutual shaping of hydrological extremes and society, there is a need for both empirical and theoretical research exploring numerous river basins,

floodplains and cities as coupled human-water systems. Figure 3 schematizes how internal feedback mechanisms within the systems consist of: i) impacts and perceptions of hydrological extremes that shape society in terms of demography, institution and governance, and ii) policies and measures implemented by society that shape hydrological extremes in terms of frequency, magnitude and spatial distribution. These internal dynamics also interact with external drivers of change operating on larger or global scales

(Figure 3), i.e. climatic and human influences outside the system (Turner et al., 2003).
One of the challenges in unravelling the interplay of hydrological extremes and society is the different time and space scales of drought and flood events. While the duration of flood events ranges from hours to days, drought has much longer lifetimes, in the order of weeks, months or even years. Similarly, spatial scales of flood events are typically smaller than those of drought conditions (Van Loon, 2015). As a result,

the integrated effects of these hydrological extremes on society and the associated feedback loops are significantly different. For instance, at level of crisis management, more time for decision making is available in the case of drought than for flood events. Also, while some flood protection measures can decided and implemented at the local level within one or few municipalities, drought policies require agreements at regional scales.

Yet, water management policies account for both hydrological extremes. Moreover, for large river basins, the periodicity or clustering of drought and flood events seem to be more coherent in time and space. This due to mass balance reasons as well as the fact that flood and drought periods are often produced by





atmospheric blocking (e.g. Francis and Vavrus, 2012). Lastly, as mentioned in the previous section, the dynamics of human impacts on flood events depend on human responses to drought events, and vice versa. Thus, in the Anthropocene, it is essential to consider both hydrological extremes.

In this context, we present a new model that mimic the interplay between water management and

hydrological extremes. This conceptualisation builds on similar efforts that were recently made in socio-hydrology (Di Baldassarre et al., 2013b; 2015; Viglione et al., 2014; Kuil et al., 2016), which modelled either drought or flood events, but not both hydrological extremes. Our model focuses on the human impact on water storage via reservoirs. As the model aims to explore emerging patterns resulting from generic mechanisms, it was not based on site specific rules of operation or optimization methods. Instead,

the model was inspired by the criticism of rational decision making and optimization made by numerous scholars following the work of the Nobel laureate Daniel Kahneman. In particular, Tversky and Kahneman (1973) formulated the availability heuristic as the bias due to the fact that decision makers estimate the probability of events not only based on robust evidence, but also "by the ease with which relevant instances come to mind". Tversky and Kahneman (1973) showed that this judgmental heuristic

leads to systematic biases. By extending this concept, we develop a stylised model that simulate the mutual shaping of hydrological extremes and water management.

The model is based on the use of a reservoir, which is used to schematically characterise changes in water storage caused by human activities (Figure 1), In particular, by considering a time series of natural river discharge ($Q_N$) as inflow, the actual river discharge ($Q$) can be derived as outflow from the variation in

time of the reservoir storage ($S$) using a mass balance equation:

$$Q = Q_N - \frac{dS}{dt} \tag{1}$$

By assuming a linear reservoir with a storage coefficient ($k$), the actual river discharge is related to the volume of water stored in the reservoir ($S$) as:

$$Q = \frac{S}{k} \tag{2}$$

To capture the typically high release of water when reservoirs are full, e.g. overflows, we assume that if the storage ($S$) is above a certain threshold ($S_{max}$), the actual river discharge will have an additional





component which is, for the sake of simplicity, linearly proportional to the difference between $S$ and $S_{max}$ with an overflow coefficient ($\alpha$):

$$Q = \frac{S}{k} + \frac{(S - S_{max})}{\alpha} \qquad (3)$$

We then use a dynamically changing storage coefficient ($k$) to explain the changing rules for reservoir operation. This storage coefficient is estimated as a weighted average between a value that allows to have

enough volume available during major flood events ($k_f$), and a different value that enables to keep enough water in the reservoir to cope with drought conditions ($k_d$):

$$k = \frac{M_f \cdot k_f + M_d \cdot k_d}{M_f + M_d} \qquad (4)$$

Equation (4) shows that the weights are given by two contrasting memories of the reservoir management system, i.e. flood memory ($M_f$) and drought memory ($M_d$), which are assumed to change over time depending of actual flow conditions:

$$\frac{dM_f}{dt} = \mu \left( \frac{Q^\beta}{Q^\beta_{N,mean}} - M_f \right) \qquad (5)$$

$$\frac{dM_d}{dt} = \mu \left( \frac{Q^\beta_{N,mean}}{Q^\beta} - M_d \right) \qquad (6)$$

Equations (5) and (6) formalize our assumption that flood memory is accumulated more than drought memory during high flow conditions ($Q > Q_{N,mean}$), while drought memory is accumulated more than flood memory during low flow conditions ($Q < Q_{N,mean}$). This assumption is inspired by the aforementioned availability heuristic (Tversky and Kahneman, 1973) and based on the empirical evidence that preparedness is very high immediately after the occurrence of extreme events that often lead to

additional pressure for changes in water management. For example, Hanak (2011)reports the decline in flood insurance coverage in California after the 1997 Central Valley flooding (Figure 4).

Equations (5) and (6) also describe that both drought and flood memories diminish exponentially over time with a decay rate $\mu$. This assumption is based on previous models of human-flood interactions (Di Baldassarre et al., 2013b; 2015; Viglione et al., 2014; Grames et al., 2016), as well as scientific work on

individual and collective memory (Anastasio et al., 2012).



The exponent $\beta$ in equations (3) and (4) is used to characterize the level of bias caused by the difference between drought and flood memories. In particular, for $\beta = 0$ both memories tend to the value of 1 over time, and $k$ becomes constant. This can be used to describe a rational decision making system whereby the proportion between $k_d$ and $k_f$ is derived with an optimal design of the reservoir to balance relative weights of drought and flood events. Increasing $\beta$, indicates increasing bias as more dynamic variations of $M_d$ and $M_f$ occur during periods of high or low flow conditions, and consequently faster changes in reservoir operation rules. As a summary, Tables 1 and 2 report the state variables and time invariant parameters, respectively, of the stylised model presented here.

To show an example of the dynamics captured by this model, we compare the results obtained with variable reservoir operation rules, which depend on the changing drought and flood memories, with the results obtained by using fixed storage coefficient to cope with either drought or flood events (Figure 5). This virtual experiment is run by solving the differential equations numerically with a finite difference method, and using flow data of the river Brisbane as input, i.e. times series of natural river discharge ($Q_N$). Figure 5 shows the shift of reservoir management and how actual river flows result from changes in operation rules. In particular, Figure 5 shows that the 2011 flood event would have had a much lower discharge if the reservoir operations aimed to cope with flood, but prolonged low flow conditions in the previous decade led to reservoir operations that enable to better cope with drought, i.e. keep more water in the reservoir instead, which lead to overflow and therefore enhanced flood levels. A plausible interpretation of the 2011 Brisbane flooding.

## 4 Conclusions and perspectives

This paper described an initial attempt to study the coevolution of water management and hydrological extremes. This is considered a first step in a broad research agenda that includes both empirical and theoretical work to uncover the mutual shaping of hydrological extremes and society (Figure 3). In particular, as described by McDonald (1989) and then discussed by Di Baldassarre et al. (2016), the development of new knowledge typically require research efforts that can be classified into five main steps: 1. Data collection and analysis; 2. Examination of these data to determine salient facts that still need a formal explanation; 3. Theory development via formulation of models capturing the salient facts;



4. Model calibration, validation and uncertainty analysis; and 5. Application of models to support the decision making process. As discussed by McDonald (1989), some scientists are deeply "engaged in work that refines or makes use of the generally accepted model" (steps 4 and 5), while others are "in the process of questioning the generally accepted model" (steps 1-3). To better understand drought and flood events in the Anthropocene, we believe that research efforts should focus on steps 1-3, since coevolutionary dynamics are still largely unknown. In particular, to develop socio-hydrological theory, there is a need for iterations between historical analyses of case studies and formal explanations of the salient facts via stylized models, such as the one presented in this paper.

Besides case studies and dynamic models, an unprecedented opportunity to explore coevolutionary dynamics across spatio-temporal scales and socio-hydrological gradients is offered nowadays by the recent proliferation in global remote sensing data and worldwide archives at relatively high spatial (between 100m and 5km) and temporal (between one day and one year) resolution. In particular, by referring to the feedback loop in Figure 3, useful sources of data include: *Hydrological extremes*: outcomes of global hydrological models; worldwide river flow archives (Hannah et al., 2011); drought and flood inundation maps derived from satellite imagery (Di Baldassarre et al., 2011); *Impacts and perceptions*: global database of damage caused by droughts and floods (EM-DAT); social media such as Twitter and Facebook; *Society*: global population data and maps of human settlements (Linard et al., 2012); satellite nightlights as proxies for economic growth and human population density (Ceola et al., 2014); *Policies and measures*: global maps of land-use, irrigation, dams and reservoirs (Bierkens, 2015); information about flood protection standards in different countries (Scussolini et al., 2016). Global comparative analyses can capitalize on this flood of data and explore whether the emerging dynamics and puzzles described in this paper are either site-specific cases that occur randomly or general patterns that emerge under specific social and hydrological conditions.

**Acknowledgements.** The present work was developed within the framework of the Panta Rhei Research Initiative of the International Association of Hydrological Sciences (IAHS), within the working groups on "Changes in Flood Risk" and "Drought in the Anthropocene". Brisbane river data were downloaded



from the Water Monitoring Information Portal of Queensland Government, Department of Natural Resources and Mines (DNRM).

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




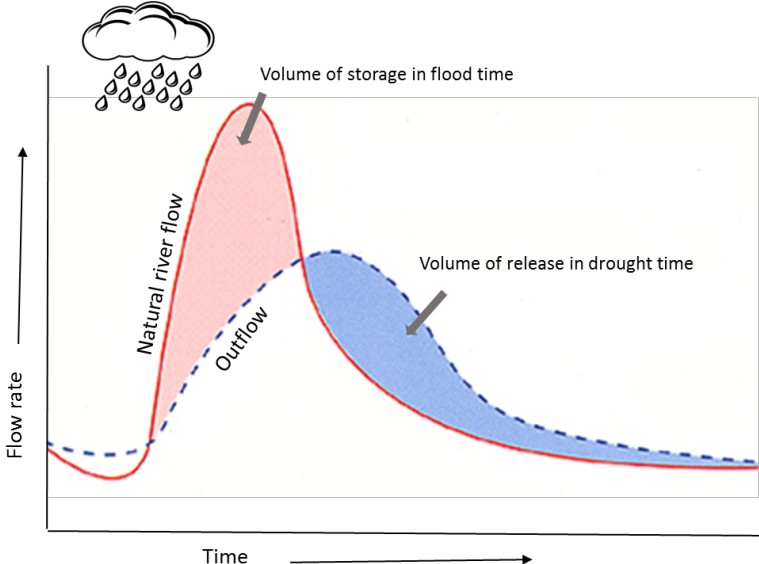

**Figure 1: Human impact on hydrological extremes. Schematic example of the impact of dams and reservoirs, which tend to mitigate both hydrological extremes, i.e. lower flows during flood events and higher flows during drought conditions.**



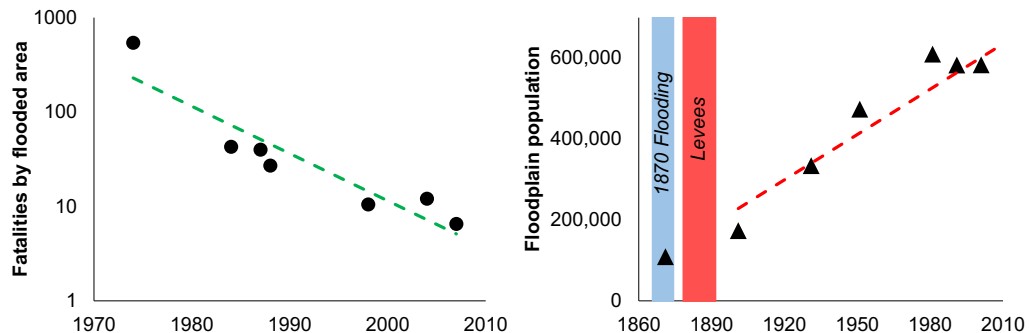

**Figure 2. Learning and forgetting effects. The left panel shows decreasing flood fatalities normalized by flooded area in Bangladesh**
5 **(data from Mechler and Bouwer, 2014), while the right panel shows increasing population in flood-prone areas in Rome (Italy),**
**following a prolonged absence of flooding due to the construction of levees (data from Ciullo et al., 2016).**





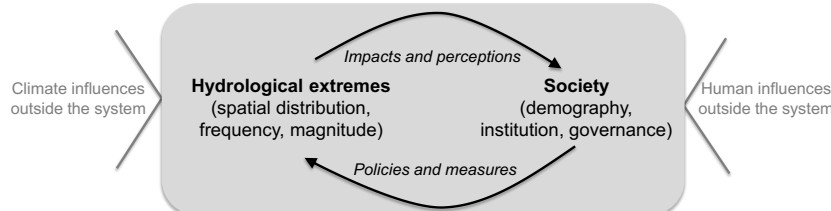

**Figure 3: Hydrological extremes in the Anthropocene. Internal feedbacks within the human-water system at the local scale (in black) and external drivers of change that operate at larger/global scale (in grey).**





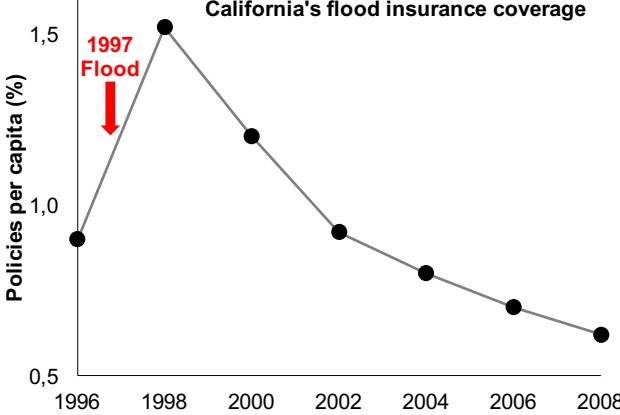

Figure 4: Changing memory and preparedness. Flood insurance coverage in California, which peaked after the 1997 Central Valley flood, and then decayed over time (data from Hanak, 2011).


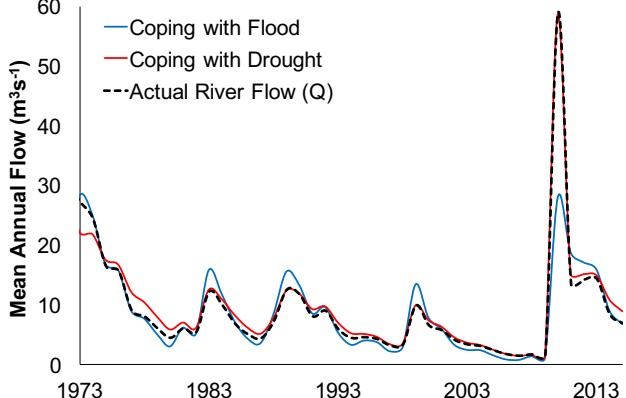

**Figure 5: Example of modelling flood and drought in the Anthropocene.** The diagram shows the result of a model run: actual river flows (black dash line) that result from changing norms in reservoir management between operation rules aiming to better cope with flood (blue line) and operation rules aiming to better cope with drought (red line).



**Table 1. Summary of time varying variables of the stylised model.**

|       | Units      | Description                  | Type  |
|-------|------------|------------------------------|-------|
| $M_f$ | [.]        | societal memory of floods    | state |
| $M_d$ | [.]        | societal memory of droughts  | state |
| $Q$   | [$L^3/T$]  | actual river flow            | state |



**Table 2. Summary of time invariant parameters of the stylised model.**

|        | Units     | Description                               |
|--------|-----------|-------------------------------------------|
| $k_f$  | [T]       | storage coefficient to cope with flood    |
| $k_d$  | [T]       | storage coefficient to cope with drought  |
| $\mu$  | [1/T]     | memory decay rate                         |
| $\alpha$ | [T]     | overflow coefficient                      |
| $\beta$ | [.]      | bias parameter                            |
| $Smax$ | [L$^3$]   | maximum reservoir storage                 |