# Peer review of "Drought and Flood in the Anthropocene: Feedback Mechanisms in Reservoir Operation"

_Earth System Dynamics, 2016_

## Referee Comment (RC1) · Anonymous Referee #1 · 13 Dec 2016

The paper by Di Baldassarre et al purports to present a stylized model of hydrological extremes and human responses. This is an exciting and important research area. However, this general framing is overly broad for the analysis performed. The actual model presented illustrates interactions between drought and flood events and human operation of reservoirs. Thus, the motivation and introduction of the paper should be more closely aligned to the model presented. Also, the model and analysis feel a bit "thin". A singular case study of "drought-then-flood" is presented for Brisbane. The authors need to show that their stylized model can capture other drought-then-flood time series, AND, importantly, that it is also able to replicate "flood-then-drought" events, as both are major motivations of the study. I suggest some major revisions that the authors may choose to undertake to make the paper more suitable for publication.

Major comments:

1.  The framing is overly broad.

    In many places, the authors claim to examine "human impacts", "human interactions", and "water management". However, the authors solely consider dam operations. The paper should be re-written to bring the motivation more in line with the actual model presented.

2.  The analysis is a bit "thin".

    a.  Fig 5 presents the main piece of analysis in the paper, which is for a drought-then-flood occurrence in Brisbane. What are the specific parameters used? How sensitive are the results to various parameter combinations? A sensitivity analysis is necessary.

    b.  What about other discharge time series?

        The study seek to understand if and how "human responses to drought events might exacerbate the impact of future floods, and vice versa" (Page 5, Lines 20-25). Yet the only case study presented explores one occurrence of a drought-then-flood in Brisbane.

        Can the model fit other similar occurrences of drought-then-flood? Are the model parameters the same? Importantly, can the model fit occurrences of flood-then-drought? What happens to the model parameters in this temporal sequence of hydrological extremes?

3.  The generic term "hydrological risk" is used, which is "broadly defined as a combination of hazard, vulnerability, and exposure".

    The natural hazard community has been trying very hard to be precise in terminology to avoid confusion and keep the external, physical driver distinct to the internal, societal vulnerability. Please be precise with terminology throughout your paper to avoid confusion.

Minor comments:

1. Throughout, terminology should be clarified to make the reader more able to understand if "reservoir inflow", "storage", or "reservoir outflow" are being considered. For example, "natural river flow" in Fig 1 is confusing and would be more clear as "reservoir inflow". Generally, it should be readily apparent if flows upstream or downstream of the reservoir are being referred to.

2. Page 9, Line 14. "Actual river flows" → "reservoir outflows"

3. Caption of Fig 1. Higher flows during drought conditions? This doesn't make sense. Please clarify.

4. Label equation variables on Fig 1. "Inflow" should be $Q\_N$ and "outflow" should be $Q$.

5. Page 5, Lines 20-25. Interesting! Can you expand upon your Melbourne time series and show more combinations of this?

6. Figure 2. Please label panels as A and B. For the right panel, do you show population time series of only flood-prone areas of Rome? Is population growth in flood-prone areas of Rome distinct to population growth in all of Rome? Importantly, is population growth shown here different to global urban population growth? Are you really just plotting the global urbanization trend?

7. Figure 4. Why do you plot policies per capita? Are you really just plotting the population growth rate in inverse here?

8. Figure 5. Please use different symbols for "coping with flood" and "coping with drought" lines, for those of us that do not print in color.

9. Table 1 and 2. Combine into a single table. Present tables with parameters used for all case studies presented (should be more than one case study in the revised version of the paper).

10. Conclusion outlines some data sources. The mismatch in spatial and temporal scale between physical and human data should be briefly mentioned and discussed.

11. The current goal of coupled human-natural (CNH) modeling is to capture feedbacks between human-natural systems, as well as internal feedbacks in human systems and internal feedbacks in natural systems. You should make it clear in your schematic (Fig 3) that you only focus on feedbacks between human-natural systems.

---

## Author Comment (AC1) · 18 Dec 2016

We would like to thank the Referee for providing constructive comments about our paper, which we believe will help us improve the description of our work. We provide here a first response to all comments and indicate the way we aim to address them during the review process.

GENERAL RESPONSE

The main issue highlighted by the Referee is that the paper is too broad in the introduction and motivation, while too narrow in the modelling part. This is due to the goal of the paper, which aims to propose a general research agenda on modelling floods and droughts in the Anthropocene, and then describe an initial attempt in that direction. Yet, the Referee has a good point and her/his comment clearly shows that there is a

discrepancy between the two parts of the paper. By going through the article again, with this criticism in mind, we eventually got to agree about this point. This is a nice aspect of peer-review and external comments. We will revise the paper by following the suggestions of the Referee, i.e. narrowing down the motivation and introduction of the paper, while expanding the modelling exercise. The next section clarifies how we aim to address all specific comments.

RESPONSE TO MAJOR COMMENTS

1. As stated above, the first part will be revised and to be more focused on reservoirs, while the modelling part will be expanded. Also, we will clarify the connection between the two parts.

2. Excellent point. We will definitely include an example of flood-then-drought as suggested by the Referee and provide more details about model parameters. Also, it is a good idea to show how key variables (e.g. k) evolve over time as flood and drought memories change.

3. According to our experience the number of definitions of risk, and each component, is immense. Even within the natural hazards community. That was the reason why we left risk broadly defined. Anyhow, following the aforementioned goal of narrowing down the introduction/motivation part of the paper, any reference to risk will just be removed as it is in fact not useful.

RESPONSE TO MINOR COMMENTS

We thank again the Referee for spotting the use of inconsistent terminology, suggesting additional discussion on a number of points, and raising technical issues with figures and captions. We will carefully revise the paper accordingly.

1-4. Indeed, current terms are confusing. The revised paper will refer to "natural inflow", "human-controlled outflow", and "reservoir storage" consistently. A specific paragraph will then be added to clarify cases in which natural inflow can be considered as representative of naturalized conditions, while human-controlled outflow can be considered as representative of actual human-modified conditions.

5. Yes, this aspect will be expanded in the revised manuscript.

6. Good point, reference to global/national/regional trends will be included to critically discuss the significance of floodplain population growth.

7. Data are given as policies per capita by Hanak et al. (2011), but the figure does not plot the population growth rate in inverse. For instance, the diagram shows that in 6 years (from 1998 to 2004) policies per capita halved (from 1,5% to about 0,75%) and California's population did not double in these 6 years. Yet, the Reviewer points to a potentially misleading information. So, the revised manuscript that will also show that in the same period policies per capita in the entire USA were essentially stable (Hanak et al., 2011). So, changes in California's policies per capita are indeed linked to the occurrence of 1997 Central Valley flood.

8-9. We'll revise figure 5 as well as table 1 and 2 as suggested.

10. Nice suggestion. A paragraph will be added to mention and discuss the mismatch in spatial and temporal scale between hydrological and social data.

11. Excellent point. A paragraph will be added to clarify that we focus on the internal feedbacks between water and human system.

---

## Referee Comment (RC2) · Anonymous Referee #2 · 30 Dec 2016

Drought and Flood in the Anthropocene: Modelling Feedback Mechanisms by Di Baldassarre et al.

This manuscript presents a framework for analyzing the feedback mechanisms of human activities and floods and droughts. The main focus is on reservoir operation and the corresponding feedbacks during droughts and floods. The framework is based on a so-called virtual model that can be used to understand the broad feedbacks between the two system (and not necessarily based on site specific rules of operation schemes). The notion of flood and drought memory, used here in the model, is really interesting and has not explored much in the past. Overall, I believe this is a good contribution and should be considered for publication after addressing the below

[Figure]

issues:

The manuscript presents an example of modelling flood and drought (Figure 5) including the actual river flows, and result from changing norms in reservoir management between operation rules aiming to better cope with flood, and operation rules aiming to better cope with drought. It would be great if the authors can plot a similar graph based on reservoir storage (i.e., observed storage, storage when the system is optimized to cope with drought, and storage when the system is optimized to cope with floods). Ideally, storage should be presented in percent of the total.

The model structure is explained well. But the parameter estimation component needs more explanation. I understand the the storage coefficient is estimated as a weighted average between a value that allows to have enough volume available during major flood events ($k_f$), and a different value that enables to keep enough water in the reservoir to cope with drought conditions ($k_d$). Please explain how $k_f$ and $k_d$ are estimated. There are other parameters in Equations 4 to 6. Are they assumed or estimated using a parameter estimation scheme?

The model uses a dynamically changing storage coefficient ($k$) to explain the changing rules for reservoir operation. Please explain the time scale of variability. Does this parameter change at monthly scale? Or seasonal?

It would be good to report the estimated parameters in Appendix or Supplementary

Materials.

Given that the model is designed to simulate long-term changes, shouldn't there be a loss term to account for direct evaporation from reservoirs? Evaporation is higher when the goal is to store water over a much longer period than when it is released faster. This may not be a big factor in overall balance. But just something to think about.

During droughts, typically, the demand is managed downstream which means the releases from reservoirs will change (i.e., a two-way feedback between demand and storage). If I understand correctly, this model does not consider this issue (?). It is worth including a brief discussion on this issue.

I suggest adding a paragraph or two on the general limitations of the model including the underlying assumptions (e.g., linearity).

There reservoir models with both constant and variable $S_{max}$ (different $S_{max}$ values for different months of a year). Is the $S_{max}$ assumed to be constant or variable here?

The focus of the model is on human impact on water storage in reservoirs. I suggest making this clear in Abstract. The current version is too broad and implies a much broader human impact assessment. Also, I suggest considering adding something like

this to the title "Feedback Mechanisms in Reservoir Operation".

---

## Author Comment (AC2) · 11 Jan 2017

We would like to thank the Referee for providing constructive comments about our paper, which we believe will help us improve the description of our work. We provide here a first response to all comments and indicate the way we aim to address them during the review process.

We welcome the Referee's suggestion to complement figure 5 with a diagram showing reservoir storage (actual one, optimized to cope with drought, and optimized to cope with floods) as a percent of the total.

We agree with the Referee that parameter estimation needs more explanation. Given the virtual nature of this numerical experiment all parameters are assumed. We will clarify this aspect in the revised manuscript and also explain the time scale of variability,

which is annual in the example provided in Figure 5, but can in principle also be monthly or seasonal (though this would include more complex reservoir operation rules).

We will follow the Referee's suggestion to report the assumed parameters as Supplementary Material or in the Tables.

The Referee makes a good point about evaporation. The model does not account for it for the sake of simplicity. Justification and discussion of the potential impact on the results will be included in the revised manuscript.

We also welcome the Referee's suggestion to include a brief discussion on the fact that during droughts, water demand is managed downstream which means that actual release from reservoirs will change. This feedback between water demand and storage is not considered in this model, but it would be an appropriate extension. This will be discussed in the revised manuscript.

We agree with the Referee about the need to add some text to discuss the general limitations of the model including the underlying assumptions, such as the use of a constant Smax.

Lastly, as discussed in our Response to Referee#1, we will clarify the focus of the model in the Abstract. Indeed, the current version is too broad. Also, we will change the second part of the title as suggested, i.e. "Feedback Mechanisms in Reservoir Operation".

---

## Editor Comment (EC1) · M. Sivapalan (Editor) · 22 Jan 2017

The paper has received useful comments and constructive criticisms from the two reviewers. The authors have responded positively to these comments. I am satisfied with the discussion and agree with the plans of the authors in terms of the revisions they propose to do. I ask them to proceed with the revisions as proposed, and when I receive the revised manuscript I will make sure it is reviewed quickly. Thank you.

---

## Author Response (AR1)

esd-2016-65
Drought and Flood in the Anthropocene: Modelling Feedback Mechanisms
Giuliano Di Baldassarre, Fabian Martinez, Zahra Kalantari, and Alberto Viglione

**RESPONSE LETTER**

We would like to thankfully acknowledge the Editor, Prof. Sivapalan, and the two Anonymous Referees for providing constructive comments about our paper. All comments were carefully considered, and we believe they helped us improve the description of our work.
We provide here a **point-by-point response**, and specify the way we addressed all comments. A **marked-up manuscript version** is reported at the end of this Response Letter.

**RESPONSE TO Reviewer#1**

**The paper by Di Baldassarre et al purports to present a stylized model of hydrological extremes and human responses. This is an exciting and important research area. However, this general framing is overly broad for the analysis performed. The actual model presented illustrates interactions between drought and flood events and human operation of reservoirs. Thus, the motivation and introduction of the paper should be more closely aligned to the model presented. Also, the model and analysis feel a bit "thin". A singular case study of "drought-thenflood" is presented for Brisbane. The authors need to show that their stylized model can capture other drought-then-flood time series, AND, importantly, that it is also able to replicate "floodthen-drought" events, as both are major motivations of the study. I suggest some major revisions that the authors may choose to undertake to make the paper more suitable for publication**

We would like to thank the Referee for providing constructive comments about our paper, which we believe will help us improve the description of our work.

One issue highlighted by the Referee is that the paper is too broad in the introduction and motivation while too narrow in the modelling exercise. This is partly due to the goal of the paper, which aims to propose a general research agenda on modelling floods and droughts in the Anthropocene, and then describe a first initial attempt in that direction.

Yet, the Referee has a point and her/his comment clearly shows that there is a discrepancy between the two parts of the paper. The paper was revised following the suggestion of the Referee. In particular, we:
i) changed the title of the paper to make it more specific (with a clear reference to "feedback mechanisms in reservoir operation" see also Response to Reviewer #2),
ii) made a more focused introduction, which also specifies better the aim (see revised Introduction), and goal of this simple model.
iii) expanding the modelling exercise and including a flood-then-drought simulation (see new figures 5b, 6a, and 6b).

MAJOR COMMENTS

**1. In many places, the authors claim to examine "human impacts", "human interactions", and "water management". However, the authors solely consider dam operations. The paper should be re-written to bring the motivation more in line with the actual model presented.**

As stated above, we change the title and revised the introduction to increase focus on reservoir operations. Previous references to risk (lines 10-15 in the original manuscript) and other human impacts, such as water abstraction, irrigation and levees (lines 20-25 in the original manuscript) were removed.

**2. The analysis is a bit "thin". a. Fig 5 presents the main piece of analysis in the paper, which is for a droughtthen-flood occurrence in Brisbane. What are the specific parameters used? How sensitive are the results to various parameter combinations? A sensitivity analysis is necessary. b. What about other discharge time series? The study seek to understand if and how "human responses to drought events might exacerbate the impact of future floods, and vice versa" (Page 5, Lines 20- 25). Yet the only case study presented explores one occurrence of a droughtthen-flood in Brisbane. Can the model fit other similar occurrences of drought-then-flood? Are the model parameters the same? Importantly, can the model fit occurrences of flood-then-drought? What happens to the model parameters in this temporal sequence of hydrological extremes?**

It is a very good point. We included an additional experiment to show a scenario of flood-then-drought as suggested by the Referee (see new Figure 6). We also included details about parameterization and initial conditions (see revised Tables 1 and 2). Also, we included new diagrams to show how the storage coefficient changes over time as a result of flood and drought memories change (see new Figure 5b and 6b).
As mentioned, we also clarified the goals of this paper of proposing a general research agenda on modelling floods and droughts in the Anthropocene, and then describing a first initial attempt in that direction (see revised Introduction in word track change). We do acknowledge that more research is needed to test more models as competing hypotheses on other case studies and via a broad sensitivity analysis.

**3. The generic term "hydrological risk" is used, which is "broadly defined as a combination of hazard, vulnerability, and exposure". The natural hazard community has been trying very hard to be precise in terminology to avoid confusion and keep the external, physical driver distinct to the internal, societal vulnerability. Please be precise with terminology throughout your paper to avoid confusion.**

According to our experience the number of definitions of risk, and each component of risk, is immense. Even within the natural hazards community.
Note that the suggested dichotomy made above between internal/societal and external/physical driver would in fact be not coherent with the methodological approach proposed here, which emphasises the two-way human-water interactions.
In other words, the external drivers of risk are for both on the physical and societal sides, likewise the internal drivers of risk are for both on the physical and societal sides (Figure 3). Anyhow, a critical reflection on the nature of risk is out of the scope of this work. Thus, we left risk broadly defined as a combination of hazard, vulnerability and exposure, which is in

fact a widely used the definition of risk (e.g. IPCC, 2014). This reference to IPCC has been added to the revised manuscript.

MINOR COMMENTS

**1. Throughout, terminology should be clarified to make the reader more able to understand if "reservoir inflow", "storage", or "reservoir outflow" are being considered. For example, "natural river flow" in Fig 1 is confusing and would be more clear as "reservoir inflow". Generally, it should be readily apparent if flows upstream or downstream of the reservoir are being referred to.**

We thank again the Referee for spotting the use of inconsistent terminology, suggesting additional discussion on a number of interesting points, and raising technical issues with figures and captions. We revised the paper accordingly.
**2,3,4 Page 9, Line 14. "Actual river flows" à "reservoir outflows" 3. Caption of Fig 1. Higher flows during drought conditions? This doesn't make sense. Please clarify. 4. Label equation variables on Fig 1. "Inflow" should be Q_N and "outflow" should be Q**

The revised paper will refer to inflow, outflow, and storage consistently. A specific paragraph will then be added to clarify cases in which inflow can be considered as representative of natural conditions while outflow can be considered as representative of actual (human-modified) conditions.

**5. Page 5, Lines 20-25. Interesting! Can you expand upon your Melbourne time series and show more combinations of this?**

We also added reference to the ongoing Oroville Dam crisis in California, which is one of the most recent disasters generated by high flow conditions that occurs immediately after prolonged droughts.

**6. Figure 2. Please label panels as A and B. For the right panel, do you show population time series of only flood-prone areas of Rome? Is population growth in flood-prone areas of Rome distinct to population growth in all of Rome? Importantly, is population growth shown here different to global urban population growth? Are you really just plotting the global urbanization trend?**

Panel were labelled A and B. It is not about global urbanisation trends: without levees all these floodplain districts in Rome could not be built as flooding would have be too frequent and made these places inhabitable. The case of Rome and the urbanization trends are in fact quite complex. This is now better explained in a recently published paper about human-flood interactions in Rome (Di Baldassarre et al., 2017). The manuscript now clarifies this point and refer to this most recent study.

**7. Figure 4. Why do you plot policies per capita? Are you really just plotting the population growth rate in inverse here?**

7. Data are given as policies per capita by Hanak et al. (2011), but the figure does not plot the population growth rate in inverse. For instance, the diagram shows that in 6 years (from 1998 to 2004) policies per capita halved (from 1,5% to about 0,75%) and California's population did not double in 6 years. Yet, we clarified this in the revised manuscript by explicitly stating (in the figure's caption) that in the same period policies per capita in the entire USA were essentially stable (Hanak et al., 2011).

**8. Figure 5. Please use different symbols for "coping with flood" and "coping with drought" lines, for those of us that do not print in color.**

Amended. This works in black and white now.

**9. Table 1 and 2. Combine into a single table. Present tables with parameters used for all case studies presented (should be more than one case study in the revised version of the paper).**
We added the initial conditions for the time-varying variables (Table 1) and the values of the time-invariant parameters used in the simulation (Table 2). Yet, we did not combine the two tables to keep a clear differentiation between time-varying variables and time-invariant parameters.

**10 Conclusion outlines some data sources. The mismatch in spatial and temporal scale between physical and human data should be briefly mentioned and discussed.**

10. Good point. A paragraph was added at the end of the conclusions to discuss the mismatch in spatial and temporal scale between hydrological and social data, as well as the need to deal with both quantitative and qualitative data. Reference to Driscoll et al. (2007) was also added for this purpose.

**11. The current goal of coupled human-natural (CNH) modeling is to capture feedbacks between human-natural systems, as well as internal feedbacks in human systems and internal feedbacks in natural systems. You should make it clear in your schematic (Fig 3) that you only focus on feedbacks between human-natural systems.**

11. The focus on the internal feedbacks between water and human system was underlined in the caption of Figure 3.

ADDITIONAL REFERENCES
Di Baldassarre, G., Saccà, S., Aronica, G. T., Grimaldi, S., Ciullo, A., and Crisci, M.: Human-flood interactions in Rome over the past 150 years, Adv. Geosci., 44, 9-13, doi:10.5194/adgeo-44-9-2017, 2017.

Driscoll, D.L., A. Appiah-Yeboah, P. Salib, and Rupert D.J.: Merging qualitative and quantitative data in mixed methods research: how to and why not. Ecological and Environmental Anthropology, 3(1), 2007.

**This manuscript presents a framework for analyzing the feedback mechanisms of human activities and floods and droughts. The main focus is on reservoir operation and the corresponding feedbacks during droughts and floods. The framework is based on a so-called virtual model that can be used to understand the broad feedbacks between the two system (and not necessarily based on site specific rules of operation schemes). The notion of flood and drought memory, used here in the model, is really interesting and has not explored much in the past. Overall, I believe this is a good contribution and should be considered for publication after addressing the below.**

We would like to thank the Referee for providing constructive comments about our paper, which we believe will help us improve the description of our work. We provide here a first response to all comments and indicate the way we aim to address them during the review process.

**The manuscript presents an example of modelling flood and drought (Figure 5) including the actual river flows, and result from changing norms in reservoir management between operation rules aiming to better cope with flood, and operation rules aiming to better cope with drought. It would be great if the authors can plot a similar graph based on reservoir storage (i.e., observed storage, storage when the system is optimized to cope with drought, and storage when the system is optimized to cope with floods). Ideally, storage should be presented in percent of the total.**

We welcome the Referee's suggestion to complement figure 5. We did not include reservoir storage though as it is linearly related to the outflow (equation 2) so there is no much information from it. We included instead the storage coefficient over time (actual one, optimized to cope with drought, and optimized to cope with floods). See new Figure 5 and 6.

**The model structure is explained well. But the parameter estimation component needs more explanation. I understand the the storage coefficient is estimated as a weighted average between a value that allows to have enough volume available during major flood events ($k_f$), and a different value that enables to keep enough water in the reservoir to cope with drought conditions ($k_d$). Please explain how $k_f$ and $k_d$ are estimated. There are other parameters in Equations 4 to 6. Are they assumed or estimated using a parameter estimation scheme?**

We agree with the Referee that parameter estimation needs more explanation. Given the virtual nature of this numerical experiment all parameters are assumed. This aspect is now clarified in the revised manuscript and the parameter values are now reported in Table 2. Also, Table 1 shows the initial conditions for the variables.

**The model uses a dynamically changing storage coefficient ($k$) to explain the changing rules for reservoir operation. Please explain the time scale of variability. Does this parameter change at monthly scale? Or seasonal?**

The time scale of variability is annual in the example provided in Figure 5, but can in principle also be monthly or seasonal (though this would include more complex reservoir operation rules). This is now explained in the text.

**It would be good to report the estimated parameters in Appendix or Supplementary**

We added them in Tables 1 and 2.

**Given that the model is designed to simulate long-term changes, shouldn't there be a loss term to account for direct evaporation from reservoirs? Evaporation is higher when the goal is to store water over a much longer period than when it is released faster. This may not be a big factor in overall balance. But just something to think about.**

The Referee makes a good point about evaporation. The model does not account for it for the sake of simplicity, but this aspect is now explicitly mentioned in the revised manuscript in describing model limitations (see below).

**During droughts, typically, the demand is managed downstream which means the releases from reservoirs will change (i.e., a two-way feedback between demand and storage). If I understand correctly, this model does not consider this issue (?). It is worth including a brief discussion on this issue.**

This feedback between water demand and storage is not explicitly considered here (see also next point), but if we understand the Referee's point correctly this aspect is at least partially (implicitly) captured by the changing storage coefficient that tends to increase during drought condition and therefore reduce outflow.

**I suggest adding a paragraph or two on the general limitations of the model including the underlying assumptions (e.g., linearity).**

We agree with the Referee and therefore added a paragraph illustrating the limitations of the model stating that "as we focus on the feedback mechanisms between flood or drought occurrence and changing reservoir operation rules, this model is highly simplified and does not account for other aspects, including the direct evaporation from the reservoir, the control of overflows (e.g. spillways), and the feedbacks between water supply and demand".

**The focus of the model is on human impact on water storage in reservoirs. I suggest making this clear in Abstract. The current version is too broad and implies a much broader human impact assessment. Also, I suggest considering adding something like C3 ESDD Interactive comment Printer-friendly version Discussion paper this to the title "Feedback Mechanisms in Reservoir Operation".**

Indeed, as discussed in our Response to Referee#1, we clarified the focus of the paper and the model in the Abstract as the original version was too broad. Also, we changed the second part of the title as suggested, i.e. "Feedback Mechanisms in Reservoir Operation".

[revised manuscript text omitted]